# Visual mode switching learned through repeated adaptation to color

Yanjun Li*, Katherine EM Tregillus, Qiongsha Luo, Stephen A Engel*

Department of Psychology, University of Minnesota, Minneapolis, United States

**Abstract** When the environment changes, vision adapts to maintain accurate perception. For repeatedly encountered environments, learning to adjust more rapidly would be beneficial, but past work remains inconclusive. We tested if the visual system can learn such visual mode switching for a strongly color-tinted environment, where adaptation causes the dominant hue to fade over time. Eleven observers wore bright red glasses for five 1-hr periods per day, for 5 days. Color adaptation was measured by asking observers to identify 'unique yellow', appearing neither reddish nor greenish. As expected, the world appeared less and less reddish during the 1-hr periods of glasses wear. Critically, across days the world also appeared significantly less reddish immediately upon donning the glasses. These results indicate that the visual system learned to rapidly adjust to the reddish environment, switching modes to stabilize color vision. Mode switching likely provides a general strategy to optimize perceptual processes.

## Introduction

When the visual system encounters different environments – for example a change in overall brightness, focus, or color – sensory processing also changes, in order to maintain accuracy and efficiency. Some of the processes producing such adjustments, called visual adaptation, unfold gradually (*Clifford et al., 2007*; *Kohn, 2007*; *Wark et al., 2007*; *Webster, 2015*). For example, putting on sunglasses can alter the color of an apple, making it difficult to determine if it is ripe, but as our visual system adapts, the apple's apparent color gradually returns to normal. For common environmental changes, it would be beneficial if the visual system could remember past adaptation, and rapidly switch to the appropriate state (*Engel et al., 2016*; *Yehezkel et al., 2010*). Such visual mode switching would aid the many functions that adaptation serves, including improving the detection or discrimination of objects and their properties (*Dragoi et al., 2002*; *Krekelberg et al., 2006*; *McDermott et al., 2010*; *Müller et al., 1999*; *Wissig et al., 2013*) and making neural codes more efficient (*Seriès et al., 2009*; *Sharpee et al., 2006*; *Wainwright, 1999*).

Empirical evidence for learning to switch visual modes is sparse and inconclusive, however. A few studies have found preliminary support for learning effects on visual adaptation (*Engel et al., 2016*; *Yehezkel et al., 2010*), but others have found little to no effect of experience (*Tregillus et al., 2016*; *Vinas et al., 2012*). Notably, previous work has not measured the consequences of moving in and out of an environment multiple times per day over many days, and none has tested for changes in the time course of adaptation with experience. Thus, it remains unclear whether people can learn to rapidly switch visual modes with experience.

Here, we used color adaptation to test for such learning: Observers wore a pair of tinted glasses, which made the world appear very reddish (the spectral transmission of the glasses as well as the monitor gamut with and without the glasses are shown in *Figure 1*). Color adaptation in such situations is relatively well-understood, and one of its main effects is that the dominant color of the environment fades over time (e.g. *Belmore and Shevell, 2008*; *de La Hire, 1694*; *Eisner and Enoch, 1982*; *Neitz et al., 2002*; *von Kries, 1902*), restoring the world to its prior, 'normal' appearance.

*For correspondence:
li000611@umn.edu (YL);
engel@umn.edu (SAE)

**Competing interests:** The authors declare that no competing interests exist.

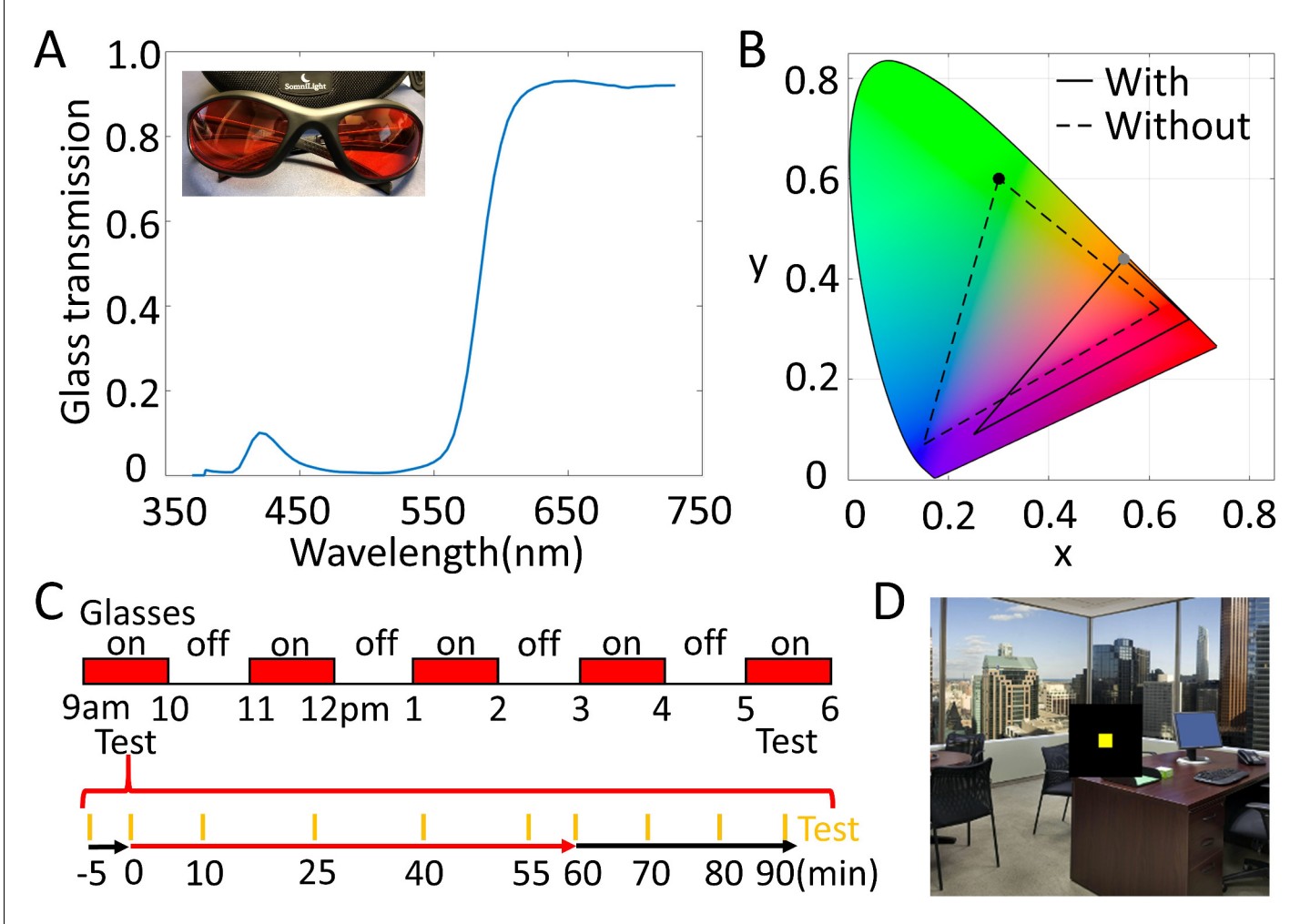

**Figure 1.** Glasses' characteristics and experimental procedures. (**A**) The red glasses used in this study and their transmission spectrum. The glasses filter out most of the energy at short wavelengths and maintain most of the energy at long wavelengths. (**B**) Monitor gamut through (solid line) and without (dashed line) the glasses plotted in CIE color space. The glasses compress the gamut and shift it toward red chromaticity. For example, the greenest light produced by the monitor (black dot) falls in an orange part of color space through the red glasses (gray dot). (**C**) Experimental procedures. The upper panel indicates the times when the observers wore the glasses within 1 day. Two test sessions were conducted, during the first and last 1 hr of wearing the glasses. The lower panel illustrates the test procedure in each session. Orange bars indicate the time of test: 5 min before putting on the glasses, immediately after putting on the glasses, then following 10, 25, 40, and 55 min of wearing the glasses. Observers then removed the glasses and were tested immediately, and 10, 20, and 30 min later. (**D**) Test display. Observers adjusted the color of a square centered on a background image of a naturalistic office environment, presented on a monitor in a fully lit room. The fixed image of the office and skyline was presented on the test display to give observers context information when making the adjustments. A black square of 5.7° separated the 0.5° square test patch from the background image. The test patch was presented for 200 ms at 1.5 s intervals, and the observer's goal was to set it to appear unique yellow. Observers viewed the test display through a 3-foot felt-lined tunnel, on a calibrated monitor, in the fully lit lab room.

Observers in the present experiment donned and removed the glasses multiple times a day for 5 consecutive days. We hypothesized that color adaptation would speed up and/or get a head start over days, such that observers would experience a much smaller perceptual change in the color of the world when they put on the red glasses, providing evidence that they had learned to switch modes. Because it may involve mechanisms beyond classical adaptation, we will use the term 'rapid adjustment' to refer to this possible empirical evidence for mode switching – that as soon as observers put the glasses on, their effects were less prominent. Different potential mechanisms behind the adjustment will be considered in the Discussion.

Observers wore the red glasses for five 1-hr periods, each separated by 1 hr without glasses (*Figure 1*). To track adaptation, we asked observers to make unique yellow settings, identifying the

wavelength of light that appears neither reddish nor greenish (*Jameson and Hurvich, 1955*). Unique yellow is a commonly used measure in color perception, in part because observers are highly consistent in their judgments (e.g. *Brainard et al., 2000*; *Jameson and Hurvich, 1955*; *Neitz et al., 2002*).

On each day, observers were tested in two sessions, once in the morning and once in the afternoon, for 5 days in a row. In each session, they performed: 1 test before putting on the glasses; 5 tests with the glasses on; and 4 tests after removing the glasses. During each test, observers made unique yellow settings for five 1-min blocks. Within each block, observers set as many matches as they could. Each datapoint in *Figures 2–5* represents the average settings across a 5-min test. The tests were all conducted in a fully lit lab room in order to provide information about the visual environment present. In a follow-up, conducted about 1 month after the main experiment, observers participated in one additional and identical testing session.

## Results

The world appeared very reddish when observers first put on the glasses, and the redness faded over time as vision adapted. *Figure 2* plots mean unique yellow settings (quantified as hue angle, see Materials and methods) as a function of time, averaging across 11 observers, for the 5 days. The

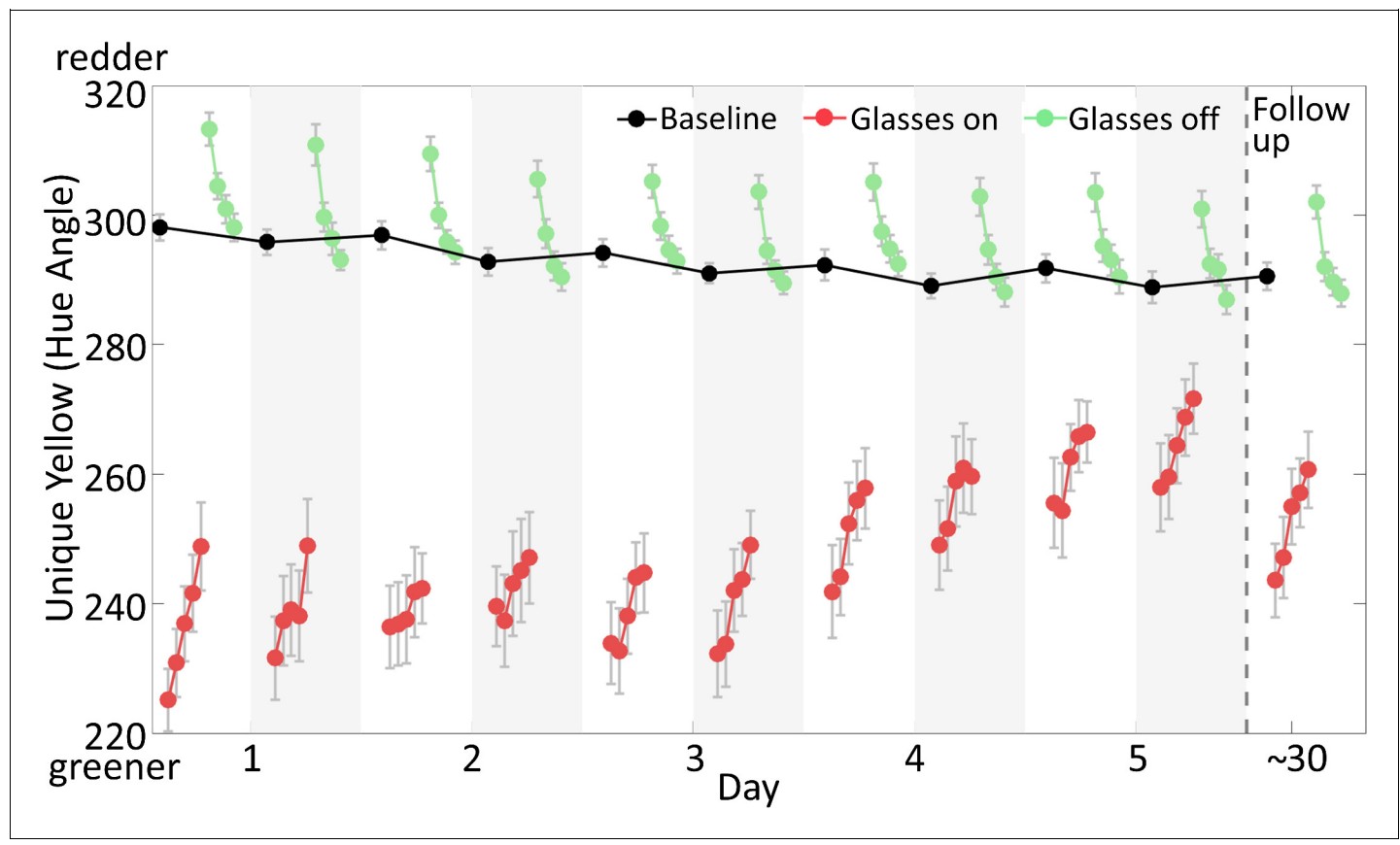

**Figure 2.** Results of the main experiment and the follow-up session. Mean unique yellow settings represented in hue angle are plotted as a function of time for 5 days and the follow-up test. The black dots are baseline settings, made at the beginning of each test session with glasses off. The white background indicates morning sessions, and the light gray background indicates afternoon. The red dots plot settings with glasses on and the green dots are settings after removing the glasses. Successive symbols are plotted for each 5-min test (see *Figure 1C*). The gray bars represent standard errors of the mean, computed across participants (N = 11).

The online version of this article includes the following source data and figure supplement(s) for figure 2:

Source data 1. *Figure 2* data.

Figure supplement 1. Baseline-corrected results of the main experiment and the follow-up session.

Figure supplement 2. Block-by-block results.

relatively small number (around 220) on the very first test with glasses on (red dots) indicates that observers' unique yellow was physically relatively green, which was required to cancel the redness produced by the glasses. The upward slope of each session's 5 settings shows that observers added less green to unique yellow over time, adapting to the red environment during the 1 hr of wearing the glasses, with the world looking less and less red. This pattern can be seen both in the morning (with white background in *Figure 2*) and the afternoon (with light gray background) session on all 5 days.

## Adjusting to the glasses became faster and stronger

Across days, observers learned to rapidly adjust to the red glasses. That is, when they first put the glasses on, the world appeared less and less reddish. This is visible in the graph by the rising trend of the first unique yellow setting in each session across days. A linear trend analysis (*Figure 3* red dots) showed that this increase was reliable ($y_t = 4.06t - 76.7 + e_t$, 95% CI [2.91, 5.21], t = 6.87, p <0.0001). A number of different mechanisms could account for this empirical observation (see

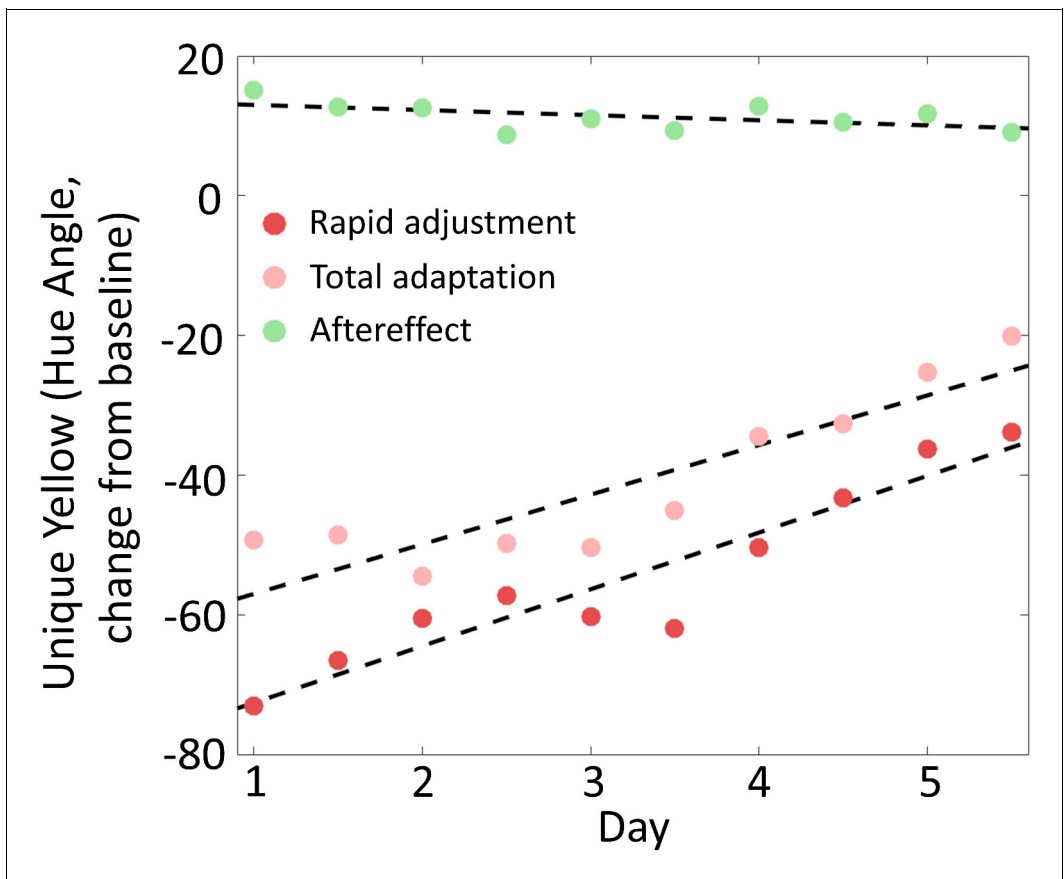

**Figure 3.** Rapid adjustment, total adaptation, and color aftereffect across 5 days. Red dots show rapid adjustments, computed as mean settings from the first 5-min test of each session with the glasses on. Total adaptation effects, denoted by the pink dots, are mean settings from the test taken after 1hr of wearing the glasses. Green dots are mean settings of the first 5-min test after removing the glasses. Data have been corrected for possible baseline shifts by subtracting the baseline value for each morning session, taken immediately before putting the glasses on. The black dashed lines are linear fits to the rapid adjustment, total adaptation, and the aftereffect. Both rapid adjustment and total adaptation effect grew significantly over days, and there was a trend for aftereffects to decrease across day.

The online version of this article includes the following source data and figure supplement(s) for figure 3:

**Source data 1.** *Figure 3* data.
**Figure supplement 1.** Rapid adjustment effects based upon the first match and the first block.

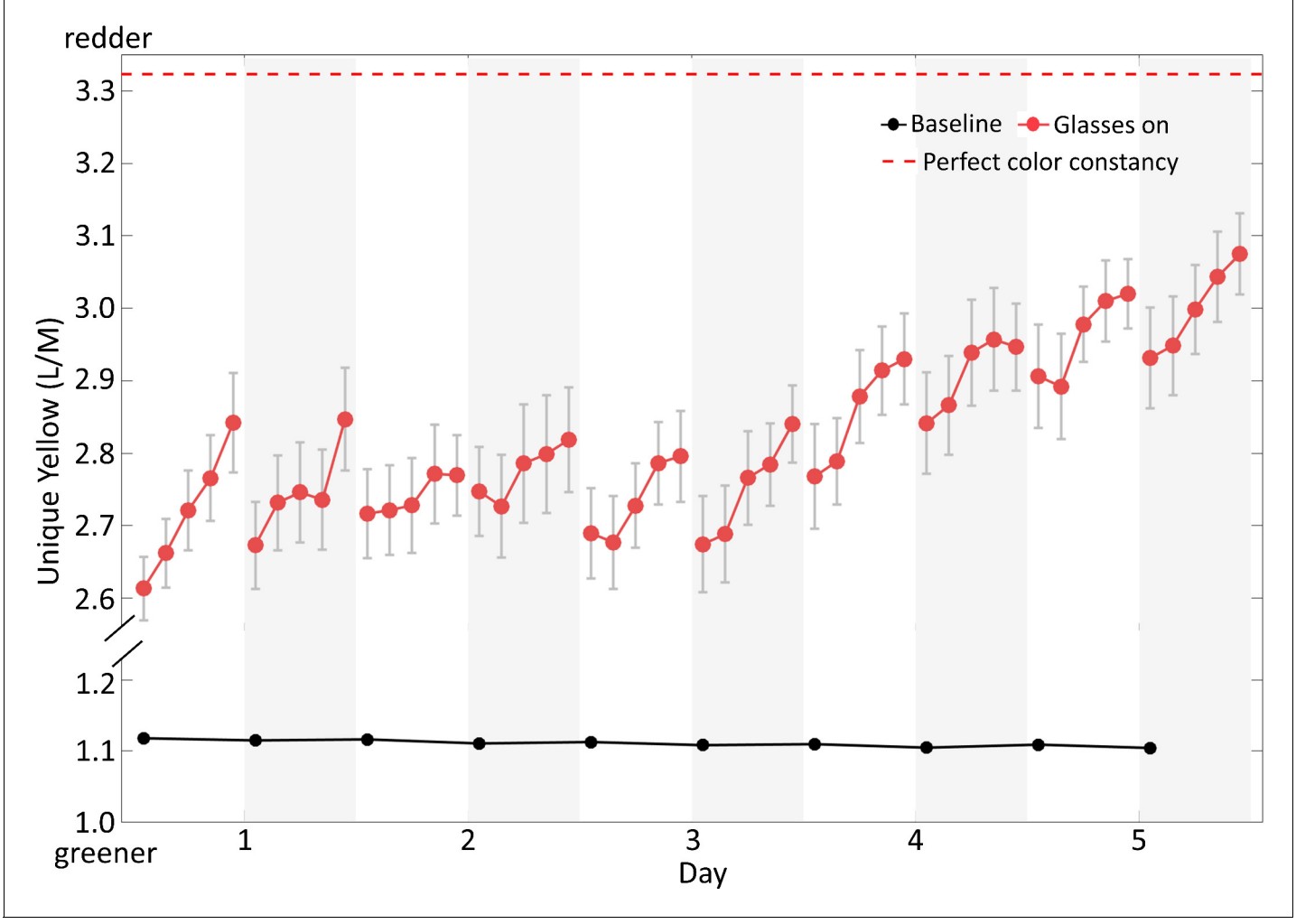

**Figure 4.** Results plotted as relative gain of cones, L/M. The red symbols show the relative gain of L and M cones (k = L/M, see text) for settings with glasses on, corrected for the red glasses transmittance. The black dots are baseline settings taken at the beginning of each test session with glasses off. If the observers showed complete absence of color constancy, the unique yellow settings with glasses on should have been at the same level as this baseline. The red dashed line above corresponds to the baseline unique yellow corrected for the red glasses' transmittance. If observers had perfect color constancy, their settings would produce identical physical colors on the monitor with and without glasses, and so should fall here when glasses were worn.

The online version of this article includes the following source data for figure 4:

**Source data 1.** *Figure 4* data.

Discussion) but the changes were not due to lingering overall adaptation across days, as baseline measurements made before putting the glasses on showed a very different trend (see below).

How rapidly did this effect arise? Each datapoint in *Figures 2* and *3* represents mean unique yellow settings averaged across the five 1-min blocks that comprised each test. To better judge the timing of effects, we repeated our analysis using observers' averaged settings within only the first 1-min block. We also repeated the analysis using observers' very first unique yellow setting in the first block. In both cases, unique yellow after donning the glasses again shifted significantly across days (t = 6.72, p<0.001 for the first block; t = 4.11, p<0.01 for the first setting), suggesting observers adjusted to the red glasses relatively quickly (*Figure 3—figure supplement 1*; *Figure 2—figure supplement 2* shows the complete time course of our results as a function of 1-min blocks. We did not have priori expectations about the subtle trends from block to block, and so leave their examination to future work).

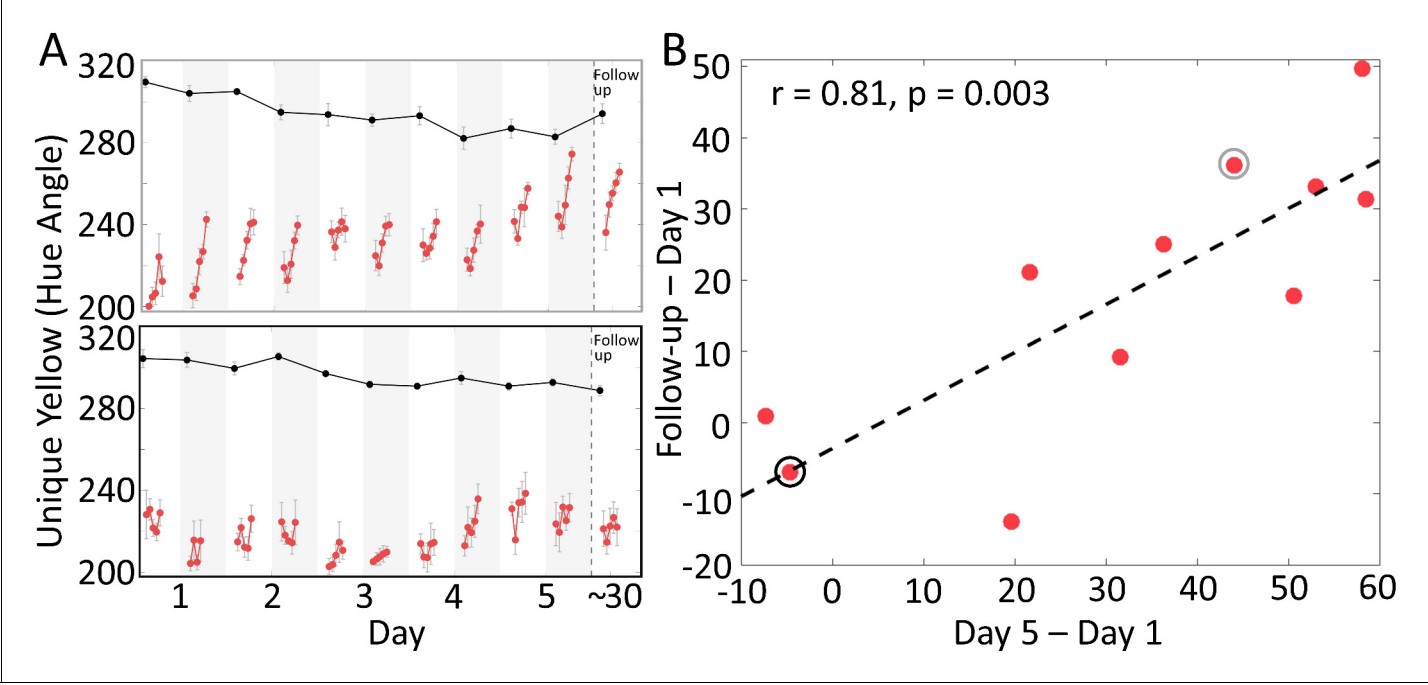

**Figure 5.** Individual differences in learning to adjust rapidly. (**A**) Complete time courses for twoobservers. One observer (upper panel) showed a gradual increase of rapid adjustment during the 5 days. This observer also retained the stronger rapid adjustment in the follow-up test. Another observer (lower panel), showed a flatter pattern across days and little effect of learning in the follow-up test. (**B**) Test-retest reliability of individual differences. The change in rapid adjustment to the glasses (relative to the 1st day) measured on the 5th day significantly correlated with the change measured in follow-up test, across observers. This indicates observers differed in their ability to learn to rapidly switch visual modes. Red dots represent observers and the dashed line is the least-square fit. The light gray and black circles denote the individuals plotted in the upper and lower portion of panel A, respectively.

The online version of this article includes the following source data for figure 5:

**Source data 1.** *Figure 5* data.

The amount of gradual adaptation to the red glasses during the 1 hr of testing, on the other hand, did not change across days. To estimate this quantity, we calculated the slope of the unique yellow settings within each 1 hr session. The grand average slope was 13.30° of hue angle toward red per hour, and there were no significant changes in slopes across test sessions (ANOVA, $F_{9,100}$ = 1.06, p=0.40). Given the increasing rapid and constant gradual effects, it is not surprising that total adaptation, the sum of the rapid and the gradual effects, quantified by the last setting with glasses on in each session, also increased across days ($y_t = 3.55t - 60.6 + e_t$, 95% CI [2.14, 4.96], t = 3.68, p < 0.01, *Figure 3*, pink dots).

## Learned mode switching was long-lasting

About 1 month (36 ± 7 days) after the main experiment, observers returned for a follow-up test. (*Figure 2*, right). Rapid adjustment to the glasses remained strong; the first test of unique yellow settings was redder than the settings from the first day of the main experiment (t = −4.83, p<0.001). However, the effect was somewhat diminished, as the follow-up settings were greener than those made on day 5 of the main experiment (t = 3.28, p<0.01). About 66% of the change across the 5 days was maintained in the follow-up test.

## A trend for color aftereffect to change across days

When observers removed the red glasses, they experienced a classical color aftereffect (*von Helmholtz, 1924*; *Krauskopf and Gegenfurtner, 1992*; *van Lier et al., 2009*), and reported the world looked slightly greenish, thus they added red to cancel out this aftereffect when making their unique yellow settings (*Figure 2*, green dots). There was a trend for the immediate aftereffect to become less strong across days, evident in the analyses of the first 5-min test, the first 1-min

block, and the first individual match setting (all p<0.1 and p>0.05; *Figure 3* green dots show the means of the first 5-min tests). We tracked the further decay of the aftereffect for half an hour after removing the glasses, as observers' settings shifted back toward baseline. The decay followed a roughly exponential shape, as previously reported for color aftereffects (*Fairchild and Lennie, 1992*; *Fairchild and Reniff, 1995*; *Wright and Parsons, 1934*). The decay constant, as measured by an exponential fit, did not change over days ($F_{9,96} = 0.01$, p=1).

## Baseline unique yellow became slightly greener across days

Baseline values of unique yellow on each day were measured as the mean setting from the first 5-min test of the morning session, made before putting the glasses on; these settings were preceded by many hours (averaging approximately 15) without glasses wear, and were made without the glasses on. We observed a small but significant shift in baseline unique yellow settings over time, visible in *Figure 2* (black dots) as the hue angle of baseline shifting toward green ($y_t = -0.94t + 298.3 + e_t$, $95\%\ CI\ [-1.30, -0.58]$, t = −3.33, p < 0.01). This is surprising because adapting to the red glasses makes redness more neutral over time, thus resulting in redder unique yellow (see Discussion).

To make sure our main finding of greater rapid adaptation did not depend upon this shift in baseline, we corrected its effect by subtracting the baseline setting in the morning test session on each day from all settings within the day. These baseline-corrected results showed a very similar overall pattern across days as the uncorrected data, although some effects became slightly larger (*Figure 2—figure supplement 1*).

## Color constancy increased across days

Color constancy, an important benefit of adaptation, is the extent to which objects appear the same color despite changes in viewing conditions (e.g. *Brainard and Radonjić, 2014*; *Foster, 2011*; *Witzel and Gegenfurtner, 2018*). Such stability against transient features of the environment allows color appearance to provide reliable information about object identity and state (e.g. the ripeness of an apple).

One definition of perfect color constancy is when the same physical entity, a surface or light source, is perceived as identical under different viewing conditions. In experiments on monitors, where experimenters only have direct access to pixel intensities, perceived surfaces are usually estimated using modeling of likely lights and surfaces. However, the use of colored glasses in our study affords us a more direct approach.

Specifically, if observers in our experiment had perfect color constancy, then the same physical pixels on the monitor, regardless of whether they were seen as surfaces or light sources (our experiment was ambiguous in this regard), should appear unique yellow both with and without the glasses, despite the glasses' dramatic effect on the spectrum of light reaching the eye. If these conditions hold then the only difference between the two unique yellow settings would be the difference in viewing conditions: That is, the same physical world (monitor pixels) would be perceived identically (i.e. unique yellow) across the two situations, a reasonable definition of perfect color constancy.

To estimate the *amount* of constancy, we characterized the physical color reaching the eye using the relative gain of the long-wavelength (L) and medium-wavelength (M) photoreceptors. This measure assumes that unique yellow settings correspond to a balancing point between the L and M cone responses, where a scale factor (gain) may be applied to responses of one of the cone classes: L = k*M. Effects of adaptation, or other plasticity, on unique yellow can be quantified by solving for k, which is equal to L/M (*Neitz et al., 2002*).

*Figure 4* plots our results using this metric and shows that color constancy improved across days. The black dots are baseline unique yellow settings before putting on the glasses; as expected, they fell around 1, where the gain of the L and M cones was equal. The red dashed line at the top of the plot reflects perfect color constancy with glasses on, calculated by assuming that the physical color corresponding to unique yellow did not change from baseline on the first day. This identical spectrum of light would of course result in very different cone absorptions with the glasses on than off, because of the glasses' effect on the light reaching the photoreceptors. On the other hand, if observers completely lacked color constancy, unique yellow settings with glasses on would simply remain at baseline values.

Across days, observers' unique yellow settings (red dots) steadily rose toward the perfect color constancy line, indicating that color constancy improved. The very first time they put on the glasses, observers showed about 68% of perfect constancy, as calculated by the ratio between (1) the Euclidean distance between baseline and the first unique yellow setting with glasses on and (2) the distance between baseline and perfect constancy. This pre-existing constancy was presumably due to the rapid adaptation that produces the color constancy we experience in most situations (e.g. *Rinner and Gegenfurtner, 2000*; *Smithson and Zaidi, 2004*; *Webster and Mollon, 1995*). The amount of constancy grew significantly as observers learned to immediately adjust to the red glasses (t = 4.60, p<0.001), and exceeded 80% on the 5th day.

### Individual differences in learning

*Figure 5* plots changes in adaptation for individual observers. Some observers showed a large increase in the amount of rapid adjustment over 5 days (sample single observer shown in upper panel in *Figure 5A*, gray circle in *Figure 5B*), while others demonstrated a flatter pattern (lower panel in *Figure 5A*, black circle in *Figure 5B*). To test if the individual differences were statistically reliable, we computed the Pearson correlation between the changes in rapid adjustment from the first day to the fifth day, and the changes from the first day to the follow-up test. This correlation was significant (r = 0.81, p=0.003, *Figure 5B*), indicating that observers who had a larger learning effect over 5 days also retained larger amounts a month later, a form of test-retest reliability. Thus, individuals appear to differ in their ability to learn to rapidly switch visual modes.

## Discussion

Through experience, observers learned to rapidly adjust to the red glasses, with the world appearing less and less reddish as soon as they put them on. In general, such rapid adjustment allows us to compensate for changes in the visual environment (e.g. *Dragoi et al., 2002*; *Krekelberg et al., 2006*; *McDermott et al., 2010*; *Müller et al., 1999*; *Wissig et al., 2013*), while also improving neural coding efficiency (e.g. *Seriès et al., 2009*; *Sharpee et al., 2006*; *Wainwright, 1999*).

In situations where different visual environments alternate frequently, like wearing and removing glasses, the visual system repeatedly readjusts itself. Our results suggest that observers can learn to make the adjustments more efficiently over time, to the point where they can adjust almost immediately upon entering the new environment. Such visual mode switching should enable people to better handle the demands of the complex and changing visual world.

### Relation to prior work

It is well accepted that color adaptation has a 'fast' and a 'slow' mechanism and involves both receptoral and postreceptoral visual processes (e.g. *Augenstein and Pugh, 1977*; *Fairchild and Reniff, 1995*; *Rinner and Gegenfurtner, 2000*). One plausible interpretation of our results depends on these well-studied mechanisms; it is possible that through practice a fast adaptation mechanism became able to produce stronger and more rapid effects. In the motor-learning literature, this possibility has been termed 'meta-learning' because it affects parameters that govern the rate of adaptation, itself a kind of learning (e.g. *Zarahn et al., 2008*). Other alternative mechanisms are possible, however, including storage, and retrieval of adapted states (e.g. *Lee and Schweighofer, 2009*). Future work will explore these and other possibilities (see also below).

Past work examining visual mode switching has produced mixed results. For example, observers who adapted to cylindrical lenses, creating a sort of astigmatism, showed fast re-adaptation in a second testing session (*Yehezkel et al., 2010*). However, clinically astigmatic observers showed little change in adaptation during 6 months following their initial prescription of corrective lenses (*Vinas et al., 2012*). Conflicting results also appeared in color perception, where in one study adapting to yellow filters produced little change in adaptation across 5 days (*Tregillus et al., 2016*), while another report showed that long-term habitual wearers of red and green lenses can adapt more rapidly than naive observers to the color changes the lenses produce (*Engel et al., 2016*). Variability in observer populations and experimental procedures may account for these mixed findings. A final bit of evidence for mode switching comes from a different paradigm, in which learning of a visual discrimination task was specific to the visual system's adaptive state, as manipulated by inducing a motion aftereffect (*McGovern et al., 2012*).

Our paradigm differed from past work in that observers adapted to very strong perceptual changes multiple times a day, and we tracked the detailed time course of adaptation in a test setting with rich cues to context (see below). Together, these factors likely produced larger changes and more reliable measurements of adaptation than observed previously. Testing whether factors such as the frequency of environmental change have an influence on the learning effect that we observed here is an important direction for future research.

Past work on long-term adaptation to colored environments, for example wearing red glasses or living under red lights continuously for part of the day, has found that adaptation grows stronger over days (*Belmore and Shevell, 2008*; *Belmore and Shevell, 2011*; *Eisner and Enoch, 1982*; *Hill and Stevenson, 1976*; *Kohler, 1963*; *Neitz et al., 2002*). However, these studies did not measure the time course of adaptation, or if observers could learn to rapidly switch between the different viewing conditions.

These past results were also highly variable, both within and between studies (*Belmore and Shevell, 2008*; *Belmore and Shevell, 2011*; *Eisner and Enoch, 1982*; *Eskew and Richters, 2008*; *Hill and Stevenson, 1976*; *Kohler, 1963*; *Neitz et al., 2002*; *Tregillus et al., 2016*), similar to the inconsistency in prior results on mode switching. One reason for this variability may be that observers were tested with little context present. For example, most tests were made in a completely darkened room, presenting only a single small test patch, making it difficult for the visual system to determine viewing conditions, and hence the appropriate adaptive state. The test setting in our experiment provided many cues that the visual system could use to tell which environment was present, that is whether the red glasses were on or off. These context cues may be necessary for mode switching to occur, although precisely which cues are important for which environments remains to be determined.

## Other results from present work

Unexpectedly, we found that the baseline unique yellow setting, made immediately prior to the introduction of the red glasses each morning, shifted toward physically more greenish across days. The shift was in the opposite direction from the color that the glasses produced and from the shift of the adaptation effect within 1 hr. A similar trend in baseline settings was also found in two previous studies (*Engel et al., 2016*; *Tregillus et al., 2016*). While we can only speculate as to the cause of this pattern, it could be due to the aftereffect following the glasses' removal. At that point, observers' judgments indicated that the world looked greenish to them, consistent with classical color aftereffects (*von Helmholtz, 1924*; *Krauskopf and Gegenfurtner, 1992*; *van Lier et al., 2009*). Adaptation across days to this greenish tint could have produced a shift in unique yellow toward green when not wearing the glasses. Long-term adaptation to aftereffects appears to be possible in other domains (*Murch and Hirsch, 1972*; *Sheth and Shimojo, 2008*).

The strengthened rapid adaptation we observed substantially improved observers' color constancy, that is the stability of perceived color despite the changes in viewing conditions (e.g. *Brainard and Radonjić, 2014*; *Foster, 2011*; *Witzel and Gegenfurtner, 2018*). Rapid adaptation, and even faster processes including 'simultaneous' local contrast, are likely major mechanisms that serve this constancy, (e.g. *Rinner and Gegenfurtner, 2000*; *Smithson and Zaidi, 2004*). A current debate in the field is whether constancy is improved for familiar, natural illuminant changes, which our visual systems may have encountered most often (*Rüttiger et al., 1999*; *Delahunt and Brainard, 2004*; *Pearce et al., 2014*; *Radonjić and Brainard, 2016*; *Weiss et al., 2017*). Our results suggest that training with repeated exposure can improve color constancy, at least for a very strong and unfamiliar illumination change. More generally, observers show some amount of color constancy, and a variety of other perceptual constancies, in most natural settings, without any training. The extent to which these forms of visual mode switching are inborn, determined during development, or learned as an adult remains under investigation (e.g. *Jameson and Hurvich, 1989*; *Sugita, 2004*; *Yang et al., 2015*).

Relatedly, the aftereffect measured immediately upon removing the red glasses shifted toward the baseline across days, implying a faster readjustment to familiar, natural conditions over time. However, this trend was relatively small, of only modest statistical reliability, and could be specific to switches from the unnatural red-glasses conditions. The small size of the effect, if real, could be because observers have already partly learned to rapidly adjust to the natural environment, which remains controversial, as mentioned above.

## Mechanisms producing more rapid adjustment

Neurally, adaptation to changes in the dominant color has effects on several sites within the retina (*Boynton and Whitten, 1970*; *Lee et al., 1999*; *Rieke and Rudd, 2009*) as well as cortical stages of color processing (e.g. *Engel and Furmanski, 2001*; *Rinner and Gegenfurtner, 2000*). One hint toward the neural locus of change in our experiment is that behavioral changes across days were not observed in adaptation within the hour of glasses wearing. This independence from classical adaptation, which partly arises early in the visual system, suggests that mode switching may arise relatively late in processing (*Rinner and Gegenfurtner, 2000*). Identifying more precisely the extent to which learning can affect these different stages of processing could be profitably addressed in the future.

Computationally, one can view adaptation as the result of an inference process, in which the visual system must determine whether the visual environment has changed (*Grzywacz and de Juan, 2003*; *Kording et al., 2007*; *Wark et al., 2009*). Through exposure to the alternating colored and uncolored environment, observers in our experiment may have learned: (1) that the red environment was more likely (i.e. it had higher prior probability); (2) to more efficiently extract evidence of the red environment (giving it a higher likelihood); (3) that the red environment was likely to persist for a long time (making it costly to not adapt); (4) to speed inference by remembering, rather than re-inferring, the past adaptive state for the red environment. All these possibilities could produce stronger immediate adjustment, and they are not mutually exclusive. Future work could determine which factors are responsible for the changes in rapid adjustment across days.

## Individual differences

What are the sources of individual differences in the ability to learn to rapidly switch between the two states? Past work has shown that observers may display very different amounts of experimentally measured color constancy, depending upon whether they were asked to make judgments of surface reflectance or of reflected light (*Arend and Reeves, 1986*; *Arend and Goldstein, 1987*; *Radonjić and Brainard, 2016*). In a given task, observers could potentially use either of these strategies. We gave specific instructions in order to limit the impact of strategy selection (see Materials and methods); however, it is still possible that some observers could be 'thinking' more or less in making their unique yellow judgments, which could be one source of the individual differences we found here. Compliance in wearing the glasses could also theoretically account for differences, but we closely monitored compliance, and failures were very few. Future work can examine whether individual differences in other aspects of color perception, or vision more generally, can account for individual differences in mode-switching.

In sum, our results demonstrate that the visual system can learn to rapidly adjust to an experienced environment. This mode switching lessens the perceptual changes produced by changing viewing conditions, which could aid a number of perceptual tasks, for example recognition of objects or materials, discrimination between similar objects or materials, as well as improved communication with other observers. Mode switching is not limited to color vision. Similar rapid re-adaptation has been reported in audition (*Hofman et al., 1998*) and sensorimotor paradigms, in which observers adapt to prisms that rotate or displace their visual field (e.g. *Redding et al., 2005*), or force fields that disturb their motor outcomes (e.g. *Wolpert and Flanagan, 2016*). Visual mode switching also resembles context-dependent learning that arises in conditioning and other memory paradigms. Mode switching may be a general solution to the problem of maintaining consistent behavior in a changing world.

## Materials and methods

### Observers

Observers included author YL and 11 members (21 to 37 years of age) of the University of Minnesota community. All had normal color vision, as assessed by the Ishihara Color Blindness Test, and normal or corrected-to-normal (using contact lenses) visual acuity. None had worn red glasses for extended periods of time prior to this study. One of the observers recruited reported that she changed her criterion for unique yellow during the study, and her data showed very large variance in baseline across days. Her data were excluded from further analysis. Experimental procedures were approved by the

University of Minnesota Institutional Review Board. All observers provided written, informed consent before the start of the study.

## Apparatus

Visual stimuli were presented on a NEC MultiSync FP2141 cathode ray tube monitor, with screen resolution of 1024*768 pixels, and a refresh rate of 85 Hz. The monitor was calibrated using a Photo Research PR655 spectroradiometer, with gun outputs linearized through look-up tables. All visual stimuli were delivered in Matlab using the psychophysical toolbox (*Brainard, 1997*). Viewing distance was maintained at 50 cm with a chinrest.

## Glasses

Observers wore a commercial pair of bright red glasses made by *SomniLight (Shawnee, KS)*. Black baffling was added on the top of the frame to prevent light from bypassing the glasses from above. The glasses filter out most of the light at short wavelengths and let pass most of the light at long wavelengths. We measured the glasses transmittance by placing the glasses in front of the spectroradiometer and recording sunlight. The spectral transmission of the glasses (*Figure 1A*) shows that the transmittance is above 90% at wavelengths over 620 nm, and less than 10% at wavelengths below 550 nm.

To characterize the effect of the glasses on our testing display, we measured the gamut of the monitor with and without the glasses. *Figure 1B* demonstrates that the gamut of the monitor seen through the glasses becomes compressed and shifts toward red chromaticity.

## Procedure

In the main experiment, observers wore the glasses for five 1-hr periods per day, for 5 consecutive days. On each day, observers came to the lab in the morning and wore the red glasses for 1 hr, while participating in a testing session. Then, they left the lab and attended to their routine everyday activities, experiencing a variety of illumination conditions. They were asked to put on the glasses again 1 hour after they took off the glasses in the lab. During the day, they wore the glasses for three 1-hr periods, each separated by 1 hr without glasses. At the end of the fourth 1-hr period without glasses, they came back to the lab for a second testing session, identical to that in the morning. *Figure 1C*, upper panel, illustrates the procedure of the experiment. In a follow-up test session conducted about 1 month after the main experiment, observers came back and performed one additional and identical testing session.

Observers completed all tests in a fully lit room (with no window), with the aim of measuring perceptual experience in a context like their natural environment while adapting to the glasses. The screen was viewed through a 3-foot felt-lined 'tunnel', so that ambient light reaching our test display was not a significant factor. Observers sat in front of the 'tunnel' with their heads positioned on a chinrest located at its entrance.

During the test sessions, observers adjusted the color of a 0.5° square centered on a background image of a naturalistic environment (an office scene). The mean luminance of the background office image was 20 candela/m$^2$. A black square of 5.7° separated the test patch from the background image (*Figure 1D*). The goal was to set the small square to unique yellow. We gave instructions "*Your task is to adjust the small patch to yellow, which contains no red nor green in it, based on the light reaching your eye. Try not to think about what the color of the patch on the screen should be*" to observers for both tests with and without the red glasses.

The small patch was presented for 200 ms at 1.5 s intervals. To make adjustments observers pressed the left and down arrow buttons to reduce redness in the patch, right and up arrow buttons to reduce greenness in the patch, and then pressed the space bar when they had set the patch to appear neither reddish nor greenish. The left and right arrow buttons were for coarse adjustments, and the up and down arrow buttons were for finer adjustments. Observers had 20 s at the most to make one single adjustment so that they did not get stuck in making one single setting and did not adapt to the test patch.

Stimuli were created using a modified version of the MacLeod-Boynton color space (*MacLeod and Boynton, 1979*), scaled and shifted so that the origin corresponds to a nominal white

point of Illuminant C and so that sensitivity is roughly equated along the two axes (*Webster et al., 2000*).

We began by computing cone responses from the stimulus spectrum using the *Smith and Pokorny, 1975* cone fundamentals scaled so that the sum of L cone and M cone responses equaled 1 and the S cone responses divided by this sum also equaled 1. We then computed initial coordinates in the MacLeod-Boynton color space as $r_{mb} = (L - M)/(L + M)$ and $b_{mb} = S/(L + M)$. Finally, we scaled and shifted these coordinates:

$$LM = (r_{mb} - .6568) \times 2168$$

$$S = (b_{mb} - .01825) \times 6210$$

where LM is the scaled red-green coordinate, and S is the scaled S-cone coordinate, 0.6568 and 0.01825 are the MacLeod-Boynton coordinates of Illuminant C, and 2168 and 6210 are constants that scale the LM and S axes so that a value of 1 is roughly equal to detection threshold (*Webster and Mollon, 1995*).

All settings fell along the nominally iso-luminant plane (defined by the LM and S axes, with luminance set to 51 candela/m$^2$) when not wearing the glasses in order to reduce brightness effects on the judgments. The photopic luminosity function we used to define nominal isoluminance was the CIE Photopic V(λ) modified by *Judd, 1951*.

In performing the unique yellow task, observers moved the stimulus along a circle in this plane. Thus, results are shown in 'Hue Angle,' where luminance and contrast (i.e. distance from the origin in the plane) were held constant. The stimuli were not adjusted for the glasses, and thus were likely not held at strictly constant luminance or contrast for judgments made while the glasses were on. The radius of the hue circle used was 80, which is a chromatic contrast of roughly 80 times detection threshold (see above) and was kept constant during the adjustment procedure.

Observers could adjust the angle of the stimulus with coarser or finer steps of 5 or 1 degree of hue angle per button press, respectively. Button presses had no effect once observers reached a green endpoint at 200° in hue angle and a red endpoint at 360° of hue angle. At the beginning of each trial, the hue angle of the stimuli was set randomly from 290 ± 45°. We tracked observers' responses and stored each step of their adjustments. Examination of these data confirmed that they were not using the red or green endpoint as an anchor for their settings (e.g. always moving to the endpoint and then moving a fixed number of steps back).

At the beginning of each test session, observers performed five 1-min blocks of this task with natural vision. Then, they put the glasses on and immediately did 5 blocks of the task again. During each block, observers made as many matches as they could and between blocks, there was a break of a few seconds. Observers were also tested after 10, 25, 40, and 55 min of wearing the glasses. Between tests observers took a short walk and/or watched videos of their choice, or texted, on a computer or their phone.

After 1 hr, observers removed the glasses and were immediately tested again. Further tests were performed 10, 20, and 30 min after removing the glasses. The full test procedure is illustrated in the lower panel of *Figure 1C*.

## Data analysis

Initial analyses averaged hue angle across tests and observers, and plotted them as a function of test time and day. In order to compare unique yellow settings with and without the glasses, we also characterized the results in terms of relative gain of the cone photoreceptors (*Neitz et al., 2002*). The analysis assumes that unique yellow settings correspond to a balancing point between the L and M cone responses, where a scale factor (gain) is applied to responses of one of the cone classes: L = k*M. Effects of adaptation can be quantified by solving for k using estimates of the cone responses to the stimulus for each unique yellow setting.

We computed relative gains of cones as follows: First, we calculated the spectra of the unique yellow settings by multiplying the RGB values of the observers' settings by the gun spectra of the monitor and summing the outputs of the three guns. For the settings made with the glasses on, we further multiplied the monitor spectra by the transmission spectrum of the glasses. The spectra of the settings were then multiplied by the cone fundamentals to compute cone absorptions, using

*Stockman and Sharpe, 2000* fundamentals, with peaks scaled to 1. Lastly, the absorptions were converted into relative gain by the ratio of L/M (which solves for k in the equation above). This same quantity was computed for settings made both with and without the glasses.

## Acknowledgements

This study was funded by NSF-BCS 1558308. We thank Victoria Papke and the many other research assistants who helped with this study. Michael A Webster and Rhea Eskew provided valuable discussion, and we thank the three reviewers.

## Additional information

### Funding

| Funder | Grant reference number | Author |
|---|---|---|
| National Science Foundation | NSF-BCS 1558308 | Stephen A Engel |

The funders had no role in study design, data collection and interpretation, or the decision to submit the work for publication.

### Author contributions

Yanjun Li, Conceptualization, Data curation, Formal analysis, Methodology, Writing - original draft, Project administration, Writing - review and editing; Katherine EM Tregillus, Conceptualization, Data curation, Writing - review and editing; Qiongsha Luo, Data curation; Stephen A Engel, Conceptualization, Supervision, Funding acquisition, Writing - review and editing

### Author ORCIDs

Yanjun Li https://orcid.org/0000-0002-0210-2305
Katherine EM Tregillus https://orcid.org/0000-0002-9543-6012
Stephen A Engel https://orcid.org/0000-0002-5241-6433

### Ethics

Human subjects: Experimental procedures were approved by the University of Minnesota Institutional Review Board (STUDY00002354). All subjects provided written, informed consent before the start of the study.

### Decision letter and Author response

Decision letter https://doi.org/10.7554/eLife.61179.sa1
Author response https://doi.org/10.7554/eLife.61179.sa2

## Additional files

### Supplementary files

• Transparent reporting form

### Data availability

Data and code for figure reconstruction have been deposited in OSF. Dataset name is VisualModeSwitchingDataset. https://osf.io/eru9g/, https://doi.org/10.17605/OSF.IO/ERU9G.

The following dataset was generated:

| Author(s) | Year | Dataset title | Dataset URL | Database and Identifier |
|---|---|---|---|---|
| Li Y | 2020 | VisualModeSwitchingDataset | https://doi.org/10.17605/OSF.IO/ERU9G | Open Science Framework, 10.17605/OSF.IO/ERU9G |

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
