## [Decision Letter]

**Acceptance summary:**

To maintain accurate perception, vision has to adapt in a changing environment. This paper provides convincing behavioral evidence that the visual system can learn to rapidly adjust to an experienced environment. Such learning may help stabilize vision perception and optimize perceptual processes. For instance, it may support enhanced color constancy across spectral environments an individual human encounters with some regularity, and aid many perceptual tasks, for example recognition of objects or materials.

**Decision letter after peer review:**

Thank you for submitting your article "Visual mode switching learned through experience" for consideration by *eLife*. Your article has been reviewed by Joshua Gold as the Senior Editor, a Reviewing Editor, and three reviewers. The following individuals involved in review of your submission have agreed to reveal their identity: David Brainard (Reviewer #1); Larry Maloney (Reviewer #3).

The reviewers have discussed the reviews with one another and the Reviewing Editor has drafted this decision to help you prepare a revised submission.

Summary:

It is very well established that chromatic adaptation occurs on short time scales (milliseconds to minutes) and reasonably well established that it occurs on longer time scales (hours to weeks). The question asked here is whether the visual system can learn, over a relatively long time scale (days), to accelerate/enhance its short-term adaptation (seconds to tens of minutes). Such learning might support enhanced color constancy across spectral environments an individual encounters with some regularity. The question is theoretically interesting and not much studied experimentally. The experiments appear to have been carefully executed and analyzed. The authors determine that there are two adaptation processes (rapid and gradual) and that the rapid process learns to anticipate. This "learning to adapt" effect allows individuals to adjust more rapidly to a change in visual experience when they have experienced a similar change previously. The effect was long lasting, still present in diminished form a month after completion of the main part of the experiment. They also found evidence for increased color constancy across days.

The experiments document the effect but do not much constrain the nature or site of the underlying mechanisms. In this sense, this study opens the door to future computational and experimental studies that attempt to dissect and explain the phenomenon.

Essential revisions:

1) The key phenomenon to explain is the decreasing onset of the rapid effect after change in illuminant condition. The idea that there are two stages of adaptation one of which can be driven by predictions of the external environment is exciting. The authors advance a probabilistic model of change detection but no details are given. There is a literature on two-stage models of adaptation (e.g., Pugh and Augenstein, 1977) that should be mentioned.

2) The authors portray their main finding sometimes as "increases in the amount of adaptation the visual system produces immediately upon putting on the glasses" or as learning "to shift rapidly to a partially adapted state". Thy do not seem to be equivalent mechanisms. The first refers to neural circuits adapting more rapidly/extensively after experience with the environment. The second suggests a learning effect-where the short timescale (e.g. seconds) adaptation is the same but the learning allows circuits a head start. It would be helpful to lay out these scenarios clearly in the text and use a consistent characterization of the main result.

3) It is interesting that the after-effect does not change with experience. Although subjects adjust more quickly to the glasses (e.g. they've learned the relevant adaptation state, Figure 2 red points), they haven't learned to undo the effect of removing the glasses (Figure 2, green points). The authors should discuss what they think about these findings.

4) We commend the authors for presenting a summary of the individual differences seen in the data, which are large. The origin of these differences is not clear, and together with mixed reports in the few studies of this sort that are reported in the literature mean some caution is required in interpretation of results. The possibility of some subjects "thinking about what they see" and responding on that basis comes to mind. This is a thorny issue that plagues most studies of constancy and is not reason to hold up publication of the present work, but it should be discussed.

5) How sure are you that subjects complied with glasses wearing instructions throughout the day? Could compliance relate to individual differences?

6) Color constancy: It is not clear what the logic is and how the color constancy calculations were done.

To make a perfect constancy prediction, one starts with some analysis of what surface reflectance corresponded to unique yellow in the glasses off condition, based on some assumption about ambient illumination and some constraints on surface reflectance functions. Then one asks what settings would correspond to that surface reflectance function under the changed illumination (here induced by the glasses). A calculation like this may have been done, but it is not provided.

Why is the proposed metric a measure of constancy? The rationale for the metric used (and claims made) is a single reference. Please clarify and help the readers understand better how the index captures constancy.

7) There are clearly a number of different ways to represent these data. Hue angle in the stimulus is fine and direct, and the L/(L+M) used in Figure 4 is also fine. Another alternative is the relative gain of the L and M cones at unique yellow, on the assumption that unique yellow represents L-kM = 0 for some k in each state of adaptation. That then gives k = L/M. Would this gain oriented expression of the data lead to more insight than the L/(L+M) version? It may be worth taking a look as it refers the data back to a hypothesized mechanistic state of the visual system. Would viewing either this or the L/(L+M) representation on an expanded within session time scale, separately for each session, reveal more about the dynamics, especially if (as suggested below in the context of Figure S2) the individual settings made once per minute in the within-session blocks were shown explicitly.

8) One aspect of the design that is likely important is that observers alternated between the two environments several times per day. It may be that frequency of environmental change is an important factor. Another factor is that the environment they were in was moderately complex, an office.

From Figure 1 it looks like the ambient environment for the experiment varied from day to day with changes in outdoor lighting coming in through the windows. Was there any characterization of this? Are the authors at all concerned that this variation may have affected results? Can they give any guidance to future researchers who would like to try to replicate their experiments as closely as possible, in terms of room size, relative amount occupied by windows, what the indoor lights were, ambient illumination level relative to display, etc.

9) Please say a little more about conversion to MB space and displayed stimuli.

a) How were peaks cone fundamentals scaled relative to each other when computing LMS, for subsequent computation of L/L+M and S/L+M. It seems that those two quantities are the Lmb and Smb passed into the computation of LM and S. Not sure S is the best choice of notation for the latter.

b) What photopic luminosity function was used to define nominal isoluminance. Given use of Stockman-Sharpe fundamentals one might infer the new CIE standard that is a weighted sum of those, or you might have used CIE 1931, or Judd-Vos, or.…

b) Please give the actual radius of the hue circle used in the adjustments, as well as the hue angle spacing for coarse and fine adjustments so that it would be possible for someone to produce your stimuli.

c) Was a full hue circle used, or were there endpoints? If end points, how confident are you that subjects didn't use those as a reference and count steps from there, or less explicitly anchor their adjustments to an estimated midpoint of the range provided? Learning of such strategies could masquerade as learning to adapt.

d) What is the luminance of the test patch that is being adjusted? Subsection “Apparatus” says the background luminance was 41.85 cd/m2, but later and in the picture, this is described/shown as black, which is surprising unless the ambient in the room was very high luminance.

10) Figure S2, top panel interpretation. The pattern of results is a little hard to interpret. We'd expect the least adaptation for the first setting, so in general these should be lower on the y-axis than the corresponding points in Figure 2. That does not appear to be the case in many instances. See the first group of glasses on settings, for example. Any comment? Are the first settings just really noisy? It might be clearer if each individual setting were plotted, rather than just providing the comparison of the first to the mean.

---

## [Author Response]

Essential revisions:1) The key phenomenon to explain is the decreasing onset of the rapid effect after change in illuminant condition. The idea that there are two stages of adaptation one of which can be driven by predictions of the external environment is exciting. The authors advance a probabilistic model of change detection but no details are given. There is a literature on two-stage models of adaptation (e.g., Pugh and Augenstein, 1977) that should be mentioned.

We agree that the manuscript failed to emphasize the past literature on two-stage models of adaptation, and our revision has corrected this oversight. We also have added some speculative discussion on mechanistic models. Specifically, we now mention the work cited above in our discussion, along with Fairchild and Reniff, (1995) and Rinner and Gegenfurtner, (2000), both of whom also propose “fast” and “slow” adaptation mechanisms.

Subsection “Relation to prior work”:

“It is well accepted that color adaptation has a ‘fast’ and a ‘slow’ mechanism and involves both receptoral and postreceptoral visual processes (e.g. Augenstein and Pugh, 1977; Fairchild and Reniff, 1995; Rinner and Gegenfurtner, 2000). One plausible interpretation of our results depends on these well-studied mechanisms; it is possible that through practice a fast adaptation mechanism became able to produce stronger and more rapid effects. In the motor-learning literature this possibility has been termed ‘meta-learning’ because it affects parameters that govern the rate of adaptation, itself a kind of learning (Zarahn et al., 2008). Other alternative mechanisms are possible, however, including storage, and retrieval of adapted states (Lee and Schweighofer, 2009). Future work will explore these and other possibilities (see also below).”

2) The authors portray their main finding sometimes as "increases in the amount of adaptation the visual system produces immediately upon putting on the glasses" or as learning "to shift rapidly to a partially adapted state". Thy do not seem to be equivalent mechanisms. The first refers to neural circuits adapting more rapidly/extensively after experience with the environment. The second suggests a learning effect-where the short timescale (e.g. seconds) adaptation is the same but the learning allows circuits a head start. It would be helpful to lay out these scenarios clearly in the text and use a consistent characterization of the main result.

The manuscript was unfortunately unclear on this very important point, and we have revised it accordingly. In the revised manuscript we now use the term "rapid adjustment" in the Introduction and Results. We have added text to the Introduction stating that this language is meant to only refer to the empirical observation that as soon as the observers put the glasses on, they were more adjusted to them and that a number of different mechanisms, such as the ones referred to above, could account for this. We have added to the Discussion that either or both of the above interpretations are consistent with our present data.

Introduction:

“We hypothesized that color adaptation would speed up and/or get a head start over days, such that observers would experience a much smaller perceptual change in the color of the world when they put on the red glasses, providing evidence that they had learned to switch modes. Because it may involve mechanisms beyond classical adaptation, we will use the term “rapid adjustment” to refer to this possible empirical evidence for mode switching -- that as soon as observers put the glasses on, their effects were less prominent. Different potential mechanisms behind the shift will be considered in the Discussion.”

Subsection “Relation to prior work”:

“One plausible interpretation of our results depends on these well-studied mechanisms; it is possible that through practice a fast adaptation mechanism became able to produce stronger and more rapid effects. In the motor-learning literature this possibility has been termed ‘meta-learning’ because it affects parameters that govern the rate of adaptation, itself a kind of learning (Zarahn et al., 2008). Other alternative mechanisms are possible, however, including storage, and retrieval of adapted states (Lee and Schweighofer, 2009). Future work will explore these and other possibilities (see also below).”

3) It is interesting that the aftereffect does not change with experience. Although subjects adjust more quickly to the glasses (e.g. they've learned the relevant adaptation state, Figure 2 red points), they haven't learned to undo the effect of removing the glasses (Figure 2, green points). The authors should discuss what they think about these findings.

This too is an important point, and we now discuss it more completely in the paper. We conducted more detailed analyses of the aftereffect, measured immediately upon removing the glasses, and found a trend for it to become less strong over days. As for our rapid adaptation to the glasses, we examined changes across days in the first 5-minute test, the first one-minute block, and the first individual match setting (see reply to comment, below). This change of the after-effect was not reliable at p < 0.05 in any of these analyses, but was p < 0.1 in all three. We now address this trend in the discussion as well, where we suggest it provides further evidence for improved mode-switching with experience. The reduced learning effect across days could be because observers already have partly learned to switch rapidly to natural viewing conditions through daily experience, or even evolution.

Subsection “Other Results from Present Work”:

“Relatedly, the after-effect measured immediately upon removing the red glasses shifted toward the baseline across days, implying a faster readjustment to familiar, natural conditions over time. However, this trend was relatively small, of only modest statistical reliability, and could be specific to switches from the unnatural red-glasses conditions. The small effect, if real, could be because observers have already partly learned to rapidly adjust to the natural environment, which remains an unresolved debate, as mentioned above.”

4) We commend the authors for presenting a summary of the individual differences seen in the data, which are large. The origin of these differences is not clear, and together with mixed reports in the few studies of this sort that are reported in the literature mean some caution is required in interpretation of results. The possibility of some subjects "thinking about what they see" and responding on that basis comes to mind. This is a thorny issue that plagues most studies of constancy and is not reason to hold up publication of the present work, but it should be discussed.

This is another important point that we have added to our Discussion. Past work has shown that observers may display different amounts of experimentally measured color constancy, depending upon whether they were asked to make judgments of surface reflectance or of reflected light (Arend and Reeves, 1986; Arend and Goldstein, 1987; Radonjić and Brainard, 2016). Observers could potentially use either of these strategies, depending upon what they were thinking during a task. In order to eliminate the impact of strategy selection, we gave instructions “Your task is to adjust the small patch to yellow, which contains no red nor green in it, based on the light reaching your eye. Try not to think about what the color of the patch should be” for both tests with and without the red glasses. However, it’s still possible that some observers who were not complying with the instructions could be "thinking" more or less in making their unique yellow judgements. We now discuss this possibility in the manuscript, as one source of the big individual differences we found here.

Subsection “Individual differences”:

“What are the sources of individual differences in the ability to learn to rapidly switch between the two states? Past work has shown that observers may display very different amounts of experimentally measured color constancy, depending upon whether they were asked to make judgments of surface reflectance or of reflected light (Arend and Reeves, 1986; Arend and Goldstein, 1987; Radonjić and Brainard, 2016). In a given task, observers could potentially use either of these strategies. We gave specific instructions in order to limit the impact of strategy selection (see Materials and methods), however, it is still possible that some observers could be "thinking" more or less in making their unique yellow judgements. This could be one source of the individual differences we found here.”

5) How sure are you that subjects complied with glasses wearing instructions throughout the day? Could compliance relate to individual differences?

We believe that compliance was good and apologize for leaving discussion of it out of the original manuscript. We sent text reminders to the observers on the first and second day about wearing the glasses during the day. In addition, every day when participants came in for the afternoon session, an RA asked them in person if they did as they were instructed in terms of glasses wearing. All reported wearing the glasses for 5 hours each day. Two participants reported having to delay wearing the glasses for ~30 minutes, one time on one day, and their testing sessions were accordingly delayed by 30 minutes to allow for full "glasses-on" time. It is possible that observers could be not reporting some lack of compliance, but we believe that this is not likely to be very frequent. We now discuss compliance in the manuscript.

Subsection “Individual differences”:

“Compliance in wearing the glasses could also theoretically account for them, but we closely monitored compliance, and failures were very few. Future work can examine whether individual differences in other aspects of color perception, or vision more generally, can account for individual differences in mode-switching.”

6) Color constancy: It is not clear what the logic is and how the color constancy calculations were done.To make a perfect constancy prediction, one starts with some analysis of what surface reflectance corresponded to unique yellow in the glasses off condition, based on some assumption about ambient illumination and some constraints on surface reflectance functions. Then one asks what settings would correspond to that surface reflectance function under the changed illumination (here induced by the glasses). A calculation like this may have been done, but it is not provided.Why is the proposed metric a measure of constancy? The rationale for the metric used (and claims made) is a single reference. Please clarify and help the readers understand better how the index captures constancy.

This suggestion is an excellent approach for studies conducted with lighting and/or surfaces simulated using a display device. Our understanding of the suggestion is that a perfect constancy prediction can be made by characterizing the observers' unique yellow settings in terms of an underlying physical stimulus, in this case the surface reflectance. The prediction of perfect constancy is that unique yellow should correspond to the same surface reflectance with and without the red glasses. Because of the use of display technology, and perhaps the observer's task of adjusting color rather than simulated reflectance, an experiment may not have direct access to the underlying surfaces and illuminants, and so whether or not there is a physical match (i.e. in reflectances) must be estimated using the modeling approach the reviewer outlines.

Sometimes, however, one does have more direct access to the underlying physics. An experiment could, for example, ask observers to choose unique yellow from a large set of physical (not simulated) Munsell papers under different physical illuminants. In this case, the perfect match prediction would be that the exact same paper (with the same reflectance) is selected under the different illuminants. In this case no modeling calculations are required for a good perfect constancy prediction.

We believe that the use of colored filters affords us this latter approach. Specifically, the perfect constancy prediction, i.e. whether the physics of a match, viewed with the red glasses on, matches the physics of the match, viewed with the red glasses off, in our case comes down to whether the physical content of the viewed scene was identical in the two viewing conditions. We do not have access to the observers' inferred reflectance functions, and indeed it is possible that they perceived our test as a light source rather than a surface, but we assume that if the light leaving the monitor was set identically with and without glasses, then a perfect match was made, because the identical physical situation in the world was perceived identically. That is, the only difference between the two unique yellow settings was that the observer was wearing the red glasses in one case – both the physical world and the perception it created (i.e. unique yellow) was constant in the two settings. To us, this seems a reasonable definition of constancy, and we apologize that it was not better explained in the manuscript. We now do so more completely.

Now, the reviewer also correctly points out that there is an additional question of what units to use to calculate how close the observer is to the perfect constancy prediction. Of course, many different units are possible, as long as they are corrected for the effects of the red glasses, and these could capture either how close the emitted light is to the perfect constancy prediction or how close the inferred surface reflectance is to that prediction. To avoid replotting our results too many times, we now choose to use the units that were suggested in comment 7) for this purpose, and have added citations supporting their use.

Subsection “Baseline unique yellow became slightly greener across days”:

“One definition of perfect color constancy is when the same physical entity, a surface or light source, is perceived as identical under different viewing conditions. In experiments on monitors, where experimenters only have direct access to pixel intensities, perceived surfaces are usually estimated using modeling of likely lights and surfaces. However, the use of colored glasses in our study affords us a more direct approach.

Specifically, if observers in our experiment had perfect color constancy, then the same physical pixels on the monitor, regardless of whether they were seen as surfaces or light sources (our experiment was ambiguous in this regard), should appear unique yellow both with and without the glasses, despite the glasses’ dramatic effect on the spectrum of light reaching the eye. If these conditions hold, then the only difference between the two unique yellow settings would be the difference in viewing conditions: That is, the same physical world (monitor pixels) would be perceived identically (i.e. unique yellow) across the two situations, a reasonable definition of perfect color constancy.

To estimate the amount of constancy, we characterized the physical color reaching the eye using the relative gain of the long-wavelength (L) and medium-wavelength (M) photoreceptors. This measure assumes that unique yellow settings correspond to a balancing point between the L and M cone responses, where a scale factor (gain) may be applied to responses of one of the cone classes: L = k*M. Effects of adaptation, or other plasticity, on unique yellow can be quantified by solving for k, which is equal to L/M (Neitz et al., 2002).”

7) There are clearly a number of different ways to represent these data. Hue angle in the stimulus is fine and direct, and the L/(L+M) used in Figure 4 is also fine. Another alternative is the relative gain of the L and M cones at unique yellow, on the assumption that unique yellow represents L-kM = 0 for some k in each state of adaptation. That then gives k = L/M. Would this gain oriented expression of the data lead to more insight than the L/(L+M) version? It may be worth taking a look as it refers the data back to a hypothesized mechanistic state of the visual system.

This is a great suggestion: We agree that the relative gain of the L and M cones at unique yellow is better at characterizing the mechanism that the weighting factors of cones are being adjusted. This has been previously used by Neitz, (2002) to reveal the plastic neural mechanism that compensates for the large variation in the ratio of L to M cones in the retina across populations. We have now replotted our Figure 4 in units of L/M. We then quantify the amount of color constancy using this unit, addressing comment 6). This is calculated by the ratio between (1) the Euclidean distance between baseline and the first unique yellow setting with glasses on and (2) the distance between baseline and the complete adaptation/perfect color constancy prediction (see above).

Would viewing either this or the L/(L+M) representation on an expanded within session time scale, separately for each session, reveal more about the dynamics, especially if (as suggested below in the context of Figure S2) the individual settings made once per minute in the within-session blocks were shown explicitly.

We agree that it is also important to look at the finer dynamics of the adaptation effects. We now make clearer in the paper that each point in our original plot represents the average settings across a 5-minute test, and that each test is in turn comprised of 5 one-minute blocks. Within each block, observers set multiple matches, and we can use those to examine the time course even more finely, which we do in response to comment 10) below.

Figure 2—figure supplement 2 plots the unique yellow settings, represented in hue angle, for each 1-minute block. To keep the plot of manageable size, we averaged the morning and afternoon session within a day, yielding 250 blocks in total. This averaging did not affect overall trends in the data. In the figure, one dot represents one block and 5 connected dots show the 5 blocks of each test. The upper section of the figure represents settings made with glasses off. The black dots are baseline settings before putting on the red glasses. The lower section of the figure shows settings made with glasses on. For reasons described in comment 10) below, the means in this figure were calculated while excluding the first match from tests 2-5, though again that did not affect the overall pattern of our results.

We include this in the supplementary material of the paper (Figure 2—figure supplement 2), replacing parts of our previous supplemental figures that also showed details of timing. Several factors complicate the interpretation of the plot: one cannot simply "connect all the dots" to see a fine-scale representation of the time course of adaptation. First, there are varying amounts of time between tests, from 5 to 15 minutes (see Figure 1C for complete testing schedule). In addition, between tests observers were viewing uncontrolled stimuli, though while keeping the glasses on or off as appropriate, that could differ substantially in color statistics from our test display (see also response to comment 10)). Finally, there was certainly adaptation to the testing display itself within a test. Accounting for these factors to make inferences about the fine-scale time course is difficult, and we plan to take up this challenge in a future publication, since the fine timing is generally an orthogonal point to our hypothesis about mode-switching. There are nevertheless interesting trends in the data, and we plan to share our complete data set for interested readers to examine. Note further that these factors do not invalidate our analyses that use just the first 1-minute block (or the first match) following donning or removal of the glasses, to characterize effects across testing sessions.

Subsection “Adjusting to the glasses became faster and stronger”:

“Figure 2—figure supplement 2 shows the complete time course of our results as a function of one-minute blocks. We did not have priori expectations about the subtle trends from block to block, and so leave their examination to future work.”

8) One aspect of the design that is likely important is that observers alternated between the two environments several times per day. It may be that frequency of environmental change is an important factor. Another factor is that the environment they were in was moderately complex, an office.From Figure 1 it looks like the ambient environment for the experiment varied from day to day with changes in outdoor lighting coming in through the windows. Was there any characterization of this? Are the authors at all concerned that this variation may have affected results? Can they give any guidance to future researchers who would like to try to replicate their experiments as closely as possible, in terms of room size, relative amount occupied by windows, what the indoor lights were, ambient illumination level relative to display, etc.

It is an important point that the frequency of environmental change might have an influence on the learning effect we observed here; we are currently conducting a follow-up experiments where subjects wore the same pairs of glasses as in the present study for 5 hours continuously per day for 5 days, producing less alternation between the two environments within each day. Increased adaptation to the glasses was also found in the 4 subjects that we have tested so far. Further analyses will be done after we collect more data, but collection has been slow due to the pandemic, and so we prefer to publish this result in a future paper. For now, we have added this as a future direction to the Discussion.

“Testing whether factors such as the frequency of environmental change have an influence on the learning effect that we observed here is an important direction for future research.”

We sincerely apologize for a misunderstanding due to our unclear description of Figure 1. Figure 1D depicts the image observers viewed on the computer screen during testing, when they were making the unique yellow adjustments. It does not depict our testing environment itself. The fixed image of the office (not a rendered scene, did not change over time) was presented on the test display to give observers context information when making the adjustments; specifically, to make it obvious to the visual system that they were wearing glasses. This might not be the case if testing was conducted while viewing a small patch on a black screen.

Our experiment was conducted in a room underground with no windows, and the ambient lighting was from stable overhead room lights. The screen was viewed through a 3-foot felt lined 'tunnel', so that ambient light reaching our test display was not a significant factor. Subjects sat in front of the tunnel with their heads located at the entrance of the tunnel. We now describe our testing room and display more clearly in the manuscript. Finally, we now also make clear that between testing sessions observers moved around freely in their everyday lives, experiencing a variety of illumination conditions.

Figure 1D legend:

“The fixed image of the office and skyline was presented on the test display to give observers context information when making the adjustments.

Observers viewed the test display through a 3-foot felt-lined tunnel, on a calibrated monitor, in a fully lit lab room.”

Subsection “Procedure”:

“Observers completed all tests in a fully lit room (with no window), with the aim of measuring perceptual experience in a context like their natural environment while adapting to the glasses. The screen was viewed through a 3-foot felt-lined ‘tunnel’, so that ambient light reaching our test display was not a significant factor. Observers sat in front of the ‘tunnel’ with their heads positioned on a chinrest located at its entrance.”

9) Please say a little more about conversion to MB space and displayed stimuli.a) How were peaks cone fundamentals scaled relative to each other when computing LMS, for subsequent computation of L/L+M and S/L+M. It seems that those two quantities are the Lmb and Smb passed into the computation of LM and S. Not sure S is the best choice of notation for the latter.b) What photopic luminosity function was used to define nominal isoluminance. Given use of Stockman-Sharpe fundamentals one might infer the new CIE standard that is a weighted sum of those, or you might have used CIE 1931, or Judd-Vos, or.…

We now provide a more detailed description of the conversion to MB space and the displayed stimuli. We have added the following information to our Materials and methods section.

Subsection “Procedure”:

“Stimuli were created using a modified version of the MacLeod-Boynton color space (MacLeod and Boynton, 1979), scaled and shifted so that the origin corresponds to a nominal white point of Illuminant C and so that sensitivity is roughly equated along the two axes (Webster et al., 2000).

We began by computing cone responses from the stimulus spectrum using the Smith and Pokorny, (1975) cone fundamentals scaled so that the sum of L cone and M cone responses equaled 1 and the S cone responses divided by this sum also equaled 1. We then computed initial coordinates in the MacLeod-Boynton color space as r_mb_ = (L-M)/(L+M) and b_mb_ = S/(L+M). Finally, we scaled and shifted these coordinates:

LM = (r_mb_ -.6568) x 2168

S = (b_mb_ –.01825) x 6210

Where LM is the scaled red-green coordinate, and S is the scaled S-cone coordinate, 0.6568 and 0.01825 are the MacLeod-Boynton coordinates of Illuminant C, and 2168 and 6210 are constants that scale the LM and S axes so that a value of 1 is roughly equal to detection threshold (Webster and Mollon, 1995).

All settings fell along the nominally iso-luminant plane (defined by the LM and S axes, with luminance set to 51 candela/m^2^) when not wearing the glasses in order to reduce brightness effects on the judgements. The photopic luminosity function we used to define nominal isoluminance was the CIE Photopic V(λ) modified by Judd (1951).”

To calculate L/M in the revised manuscript (and L/(L+M) in the original) we used the Stockman and Sharpe, (2000) cone fundamentals, with the peaks of cone fundamentals scaled to 1.

b) Please give the actual radius of the hue circle used in the adjustments, as well as the hue angle spacing for coarse and fine adjustments so that it would be possible for someone to produce your stimuli.

The radius of the hue circle used in this study was 80, which is the chromatic contrast. This contrast is at a nominal threshold and was kept constant during the adjustment procedure. The coarser and finer steps of adjustment were +/- 5 and +/- 1 degree respectively per button press. We now report these parameters in our Materials and methods section.

Subsection “Procedure”:

“The radius of the hue circle used was 80, which is a chromatic contrast of roughly 80 times detection threshold (see above) and was kept constant during the adjustment procedure.

Observers could adjust the angle of the stimulus with coarser or finer steps of 5 or 1 degree of hue angle respectively per button press.”

c) Was a full hue circle used, or were there endpoints? If end points, how confident are you that subjects didn't use those as a reference and count steps from there, or less explicitly anchor their adjustments to an estimated midpoint of the range provided? Learning of such strategies could masquerade as learning to adapt.

This also is an important point that we now add in the revised manuscript. We did not use the full hue circle, and we had a green endpoint at 200 degrees in hue angle and a red endpoint at 360 degrees of hue angle. At the beginning of each trial, the hue angle of the stimuli was set randomly from 290±45 degrees. We tracked observers’ responses and stored each step of their adjustments. Only a few observers in some blocks at the beginning of the experiment hit the green or red limit, making them highly unlikely to be used as an anchor point.

Subsection “Procedure”:

“Button presses had no effect once observers reached a green endpoint at 200 degrees in hue angle and a red endpoint at 360 degrees of hue angle. At the beginning of each trial, the hue angle of the stimuli was set randomly from 290±45 degrees. We tracked observers’ responses and stored each step of their adjustments. Examination of these data confirmed that they were not using the red or green endpoint as an anchor for their settings (e.g. always moving to the endpoint and then moving a fixed number of steps back).”

d) What is the luminance of the test patch that is being adjusted? Subsection “Apparatus” says the background luminance was 41.85 cd/m2, but later and in the picture, this is described/shown as black, which is surprising unless the ambient in the room was very high luminance.

We apologize for not being clear on the description of our test display and the misunderstanding caused by Figure 1D. As noted above, Figure 1D shows what was presented on the monitor to the observers when they were making the unique yellow settings. The office scene is not the room that observers were tested in; it is the background image displayed on the monitor and that was for giving observers context while they were making adjustments. The black square was 5.7 degrees of visual angle, and it separates the 0.5-degree test patch from the office image (the background). The background luminance was 20 cd/m2, and the luminance of the test patch was 51 cd/m2. We now have added this information to the paper.

Subsection “Procedure”:

“The mean luminance of the background office image was 20 candela/m^2^.”

“All settings fell along the nominally iso-luminant plane (defined by the LM and S axes, with luminance set to 51 candela/m^2^) when not wearing the glasses in order to reduce brightness effects on the judgments.”

10) Figure S2, top panel interpretation. The pattern of results is a little hard to interpret. We'd expect the least adaptation for the first setting, so in general these should be lower on the y-axis than the corresponding points in Figure 2. That does not appear to be the case in many instances. See the first group of glasses on settings, for example. Any comment? Are the first settings just really noisy? It might be clearer if each individual setting were plotted, rather than just providing the comparison of the first to the mean.

We thank the reviewer for the observation and agree that the first unique yellow setting made with glasses on in each test would be expected to be greener/lower on the y-axis than the subsequent matches. So, we examined the first match in the first block of each test (to define terms: each five-minute test was comprised of 5 one-minute blocks; each morning or afternoon session had 5 tests with glasses on, separated by 5-10 minutes according to the schedule shown in manuscript Figure 1C). We found that with glasses on, the first match in the first block in all but the first test in each session was unexpectedly larger in hue angle (less green) than the following matches in that block. But the first match in the first block of the first test immediately after putting on the glasses did not differ from other matches.

This pattern is shown in Author response image 1 which plots the first eight individual matches in each of the 5 blocks averaged across days. The linked red dots are the first matches in each block linked across tests, and the rest of the colored lines are the subsequent 2-8 matches in each block. The first matches in the first block of tests 2-5 made with glasses on were much higher on the y-axis than other matches (i.e. red lines in tests 2-5 for block 1 are outliers).

This pattern has a simple explanation: tests 2-5 were conducted after the 5 -10 minutes break, during which observers watched videos or texted on their phones or laptop computers. The first test, however, immediately followed a 'baseline' test with glasses off, where observers were viewing our test display. According to most observers, the screen of the phones or the computers they viewed during the break looked redder than other objects in the environment when they wore the glasses, probably because they had higher luminance than other indoor objects. This was particularly true for white backgrounds while texting. So, after viewing their phone/laptop screen for a period of time, observers likely became more adapted to reddishness than after viewing our test display. Then when observers transitioned from their phone/computer to the test display, they were in a more adapted state, so they set the first unique yellow match to be redder than other matches, which were made after looking at the test display for a while. This effect was not present on the first match in the first block of the first test upon putting on the glasses because there was no transition between different displays before the first test (which was conducted immediately after a baseline test).

Because these 4 points (first match of the first block with glasses on in test 2-5) were apparently taken under different conditions, we removed them when computing the block averages in Figure 2—figure supplement 2. Because our main results figures averaged across all blocks in a test, they were not affected by inclusion of these 4 points to a noticeable extent, and so we have left them unchanged (though we made sure all statistical trends were maintained if the outliers were removed). Note also that, figures that plot only the first test with glasses on (i.e. the scatter plots in Figure 3, and Figure 3—figure supplement 1) are unaffected by these outliers as well, because of the immediate transition from the baseline test. However, our previous plot of the entire time course using the first match was, as the reviewer noted, distorted, as we now know, by these outlier points, and so we have removed it from the manuscript.

**Author response image 1. sa2fig1:**